# Effect of Organic Matter Components on the Mechanical Properties of Cemented Soil

**DOI:** 10.3390/ma16175889

**Published:** 2023-08-28

**Authors:** Li Shao, Zhixuan Ding, Suran Wang, Kuashi Pan, Chuxue Hu

**Affiliations:** 1School of Environment and Architecture, University of Shanghai for Science and Technology, Shanghai 200093, China; shaoli1999@usst.edu.cn (L.S.); wangsuran@usst.edu.cn (S.W.); pankuashi@163.com (K.P.); 222271925@st.usst.edu.cn (C.H.); 2School of Intelligent Emergency Management, University of Shanghai for Science and Technology, Shanghai 200093, China

**Keywords:** organic matter, cemented soil, unconfined compressive strength, resistivity, micro-mechanism

## Abstract

The organic matter in soft clay tends to affect the properties of cement-stabilized soil. The influence degree of different organic matter varies. In this paper, the influence weights and mechanism of the main organic matter components fulvic acid and humic acid on the mechanical properties of cemented soil were investigated. Impacts of FA/HA (fulvic acid/humic acid) values and curing time on the unconfined compressive strength, deformation characteristics, and microstructure of cemented soil were explored through the unconfined compressive strength test and electrical resistivity test. The results show that with the increase of FA/HA, the unconfined compressive strength of cemented soil gradually decreased and the plastic properties enhanced. The increase in curing time changed the stress-strain relationship of cemented soil, and some specimens showed brittle damage. The initial resistivity and structural property parameters of cemented soil gradually decreased with the increasing FA/HA value and increased with the increase of curing time. It revealed the influence law of FA/HA and curing time change on the microstructure of cemented soil. Based on the experimental results, the quantitative relationship equations between FA/HA and curing time and unconfined compressive strength, failure strain, deformation modulus, and resistivity were established.

## 1. Introduction

Soft clay soils are commonly found in coastal and lowland areas. These soils are characterized by high water content, high compressibility, low strength, etc. Owing to these inherent poor engineering properties [1], it is often necessary to employ soil stabilization techniques to reinforce them when soft clay soils are used as foundation soils, to meet the specific requirements of engineering projects. Cement, a commonly used and stable chemical stabilizer, is widely applied for soil improvement [2,3,4]. Moreover, the soil mixing pile method holds significant development potential due to its convenient construction and lack of adverse effects on adjacent buildings [5]. However, in practical projects, it has been observed that the organic matter present in the soil has a certain influence on the formation process of cemented soil piles, consequently leading to engineering quality issues.

Organic matter in soft clay is one of the important factors affecting its engineering properties, which is mainly composed of fulvic acid and humic acid [6]. In order to better analyze the influence law and action mechanism of organic matter on the basic properties of cemented clay, many scholars have conducted a lot of research. Liang et al. [7] found that when the mass component of organic matter was less than the limiting value of 5%, the presence of organic matter seriously hindered the hydration reaction of cement, and the strength of organic matter-containing cemented soil was much lower than that of ordinary cemented soil. Liu et al. [8] used the method of freeze-thaw cycles to simulate and study the engineering properties of organic matter soil in the modified seasonal freezing zone and found that the compressive strength increased with age in a logarithmic law, and the increase in the number of freeze-thaw cycles made the strength decrease and the porosity increase. Wang et al. [9] used the limiting water content and the compressive strength as the evaluation indexes, investigated the influence law of the change of the content of the organic matter on the characteristics of the cement curing dredged sludge, and analyzed the mechanism of its action through microscopic experiments such as scanning electron microscopy. Cao et al. [10] investigated the effect of organic matter content on the strength of cemented soil. As a result, the strength of cemented clay decreased with the increase of organic matter content, and based on this test and past data, a strength model suitable for predicting the strength of cemented soil with different organic matter contents was established. Engineering practice has found that cemented soft soil is affected by the humic acid component of the soil, while related studies have shown that the effect of the fulvic acid component on the strength of cemented soils is stronger [11]. The stronger weakening effect of fulvic acid than humic acid is due to the higher content of oxygen-containing functional groups in fulvic acid, and therefore fulvic acid is more capable of binding calcium and aluminum during cement hydration, thus weakening the strength of stabilized soil [12,13]. While the current methods for studying the impact mechanism of organic matter on cemented soil are mostly difficult to operate and expensive. Therefore, using highly operational and economical detection techniques to investigate the mechanism of organic matter is currently a research hotspot.

Resistivity is a basic parameter that characterizes the electrical conductivity of a substance. The resistivity of soil mainly depends on some important structural parameters of soil, such as porosity, pore shape, pore structure, saturation, pore liquid resistivity, solid particle composition, particle shape, particle orientation, and cementation state [14]. In general, soil resistivity consists of the resistivity of soil particles and pore water [15]. Resistivity has been widely emphasized by scholars in the field of geotechnical engineering. Archie [16] established the relationship between resistivity and soil microstructure and also proposed the concept of Formation Factor and the classical resistivity formula applicable to pure argillaceous sandstone. Zha et al. [17] further verified the feasibility of the resistivity method to reveal the microstructural mechanism of stabilization/solidification of heavy metal contaminated soils through experiments and established a normalized prediction formula for the unconfined compressive strength of stabilized heavy metal contaminated soils based on resistivity. Liu et al. [18] investigated the relationships between electrical resistivity and UCS and leached Pb concentration by using a regression analysis. Zhao et al. [19] developed and validated a mechanical damage model based on resistivity that can accurately characterize the mechanical behavior of S-RM under triaxial shear. Yousuf et al. [20] demonstrated that non-contact resistivity measurements are a sensitive technique for monitoring cement hydration and microstructural development. The above studies indicate that resistivity measurement technology is a better choice for investigating the microstructure of soil. At the same time, scholars usually predict other indicators of soil based on the results of resistivity tests and propose relevant formulas.

Existing studies have established a clear link between organic matter and the mechanical properties of cemented soil. However, the relationship between the main components of organic matter, mechanical properties, deformation characteristics of cemented soil, and their underlying microscopic mechanism has been rarely studied, and quantitative analysis is lacking. Considering the resistivity method’s superiority in studying the microstructure of stabilized soil, this paper aims to adopt this method to elucidate the microscopic mechanism by which the main organic matter components influence the mechanical and deformation properties of cemented soil. Organic substances, including fulvic acid and humic acid, were incorporated into the test soil. The mixing ratio of fulvic acid and humic acid (hereinafter referred to as FA/HA, Fulvic Acid/Humic Acid) was varied to investigate their respective impacts on the mechanical and deformation properties of cemented soil through indoor experiments. Quantitative relationship equations between FA/HA and mechanical indices, electrical resistivity, and structural properties were established.

## 2. Materials and Methods

### 2.1. Test Materials

The test soil sample was obtained from the coastal phase soft soil subdistrict in Jiangsu Province. It is characterized as mucky clay with a grayish-black appearance, and its key physical indices are presented in Table 1. The measured data indicates that the soil is high liquid limit clay.

The organic matter consists of fulvic acid (FA) and humic acid (HA). Fulvic acid appears as a brownish-yellow powder, while humic acid appears as a black powder. The effective content of both components is 85%, and the influence of impurities is not considered in this test. The cement used in the test is ordinary Portland cement (P·O 42.5), and the main chemical composition measured by the XRF test is shown in Table 2.

### 2.2. Experimental Design and Sample Preparation

With a fixed organic matter content of 10%, the FA/HA values were set as 0.01, 0.25, 0.5, 2, 4, and ∞, respectively. The organic matter was mixed with the test soil and a predetermined amount of cement and water to prepare the samples. According to the test programs, unconfined compressive strength tests and resistivity tests were conducted on the specimens of identical organic matter content, cement content, moisture content, and different FA/HA with specified curing conditions. The detailed test program and parameters are provided in Table 3. 

The test soil was naturally dried, mechanically milled, and passed through a 2 mm sieve. Organic matter solutions with different FA/HA were prepared in advance for use. The dry soil was thoroughly mixed with the organic solution and cement. The mixture was then divided into three layers and placed into molds with a diameter of 50 mm and a height of 100 mm, as shown in Figure 1. After preparation, the specimens were sealed and placed in a standard curing room at a constant temperature of 20 ± 3 °C and a relative humidity of 100%. After 24 h of initial curing, the molds were removed, and the specimens were further cured in the standard curing room of the same curing condition until reaching the designated curing time.

### 2.3. Unconfined Compressive Strength Test

After the specified curing period, the unconfined compressive strength (UCS) test was conducted on the samples. The UCS test was performed using the YSH-2 lime soil unconfined pressure gauge, applying a stress ring coefficient of 21 N/0.01 mm and a loading rate of 1.0 mm/min. The test could be terminated when the force meter reading reached its peak value or became stable, with further axial deformation of 3% to 5%. If no stable reading was obtained, the test should be continued until the axial deformation reaches 20%. The average value of three sets of parallel samples was considered as the final compressive strength of the cemented soil, ensuring that the error between the parallel samples did not exceed 15%.

### 2.4. Resistivity Test

The resistivity of soil samples was measured using a resistance tester developed by the Geotechnical Engineering Research Institute of Southeast University, as depicted in Figure 2. This instrument is based on the electrical model of soil, and it measures the resistance of soil by adjusting the balance of the low-frequency current bridge. Subsequently, the resistivity of soil samples can be calculated. When measuring the unconfined compressive strength of soil samples, a plastic cushion block is inserted between the sample and the pressure-bearing plate. Electrode plates are installed both above and below the sample, and a pressure of 0.1 kPa is applied to ensure full contact between the electrode plates and the sample. The initial resistivity of the cemented soil sample is measured, which reflects the pore water and particles within the soil. Assuming that the resistivity of the soil particles and pore water does not vary much and is taken as a fixed value, the change in resistivity reflects the change in the structure of the soil. 

## 3. Results

### 3.1. Uniaxial Strength and Deformation Characteristics

#### 3.1.1. Unconfined Compressive Strength

Figure 3 illustrates the relationship between the unconfined compressive strength of cemented soil at different curing times and the variation of FA/HA. As can be seen from the figure, when the organic matter content was certain, the unconfined compressive strength of the cemented soil samples at the same curing time decreased gradually with the increase of FA/HA. The unconfined compressive strength of samples with the same FA/HA increased with the curing time. When FA/HA increased, the rate and magnitude of strength growth gradually decreased, and the strength increased more slowly at the later stage. The increase of FA/HA can lead to the reduction of cemented soil strength. Therefore, when the content of fulvic acid in the soil is too high, it should be reduced appropriately, so that the cemented soil can reach the specified strength.

The change in strength of organic cemented soil with FA/HA can be attributed to the following reasons: the acidic organic matter can bind to calcium and aluminum, neutralizing the alkaline substances produced by the hydration reaction of the curing agent, which reduces the solution’s pH and hinders the hydration reaction of the curing agent [21,22,23]. Humic acid and fulvic acid are described as coiled long-chain molecules with high molecular weights (4360–215,000 g/mol and 545-1840 g/mol respectively), with functional group densities mainly consisting of carboxyl (*COO^−^*) (4.2 and 6.2 mmol/g respectively) and phenolic (*OH^−^*) groups [12]. The related study has shown that the total acidity, carboxyl groups, and phenolic hydroxyl groups of fulvic acid are higher than those of humic acid [24]. Compared to humic acid, fulvic acid has a higher content of oxygen-containing functional groups, giving it a stronger ability to bind calcium and aluminum during cement hydration [8]. As a result, it exerts a more pronounced weakening effect on the cement hydration process and products. Additionally, the special structural characteristics of fulvic acid will hinder the formation of cement hydration products [25]. At the beginning of the hydration reaction, the higher water content in the soil intensifies the reaction, leading to more hydration products and faster strength growth. However, as the curing time increases, the reaction slows down due to water consumption, leading to a gradual reduction in the rate of strength growth [8].

The relationship between the unconfined compressive strength and FA/HA and curing time was obtained by fitting Equation (1) to the data derived from the test through regression analysis. The correlation coefficient was found to be 0.9898.
(1)$$q=0.045\cdot T^{0.47}+0.328\cdot T^{0.345}\cdot \;0.595{}^{\gamma }$$
where *γ* is FA/HA, *T* is the curing time (d), and *T* = 7 d~120 d.

Equation (1) shows that the effect of curing time on the growth of unconfined compressive strength under the influence of FA/HA can be divided into two parts: one part exhibits fixed strength growth with curing time, while the other part shows non-fixed strength growth with curing time as FA/HA increases, specifically manifesting as a reduction in the rate of strength increase.

#### 3.1.2. Stress-Strain Relationship

Figure 4 depicts the stress-strain curves of specimens at different curing times with varying FA/HA. The figure reveals that an increase in FA/HA resulted in a gradual decrease in the peak strength of the cemented soil, weakening the strain-softening characteristics and enhancing plasticity. Larger FA/HA ratios caused the short-aged cemented soil to exhibit pronounced plastic behavior (Figure 4a). Therefore, in actual projects, if the FA/HA value of the soil is detected to be relatively large, it should be appropriate to extend the curing time of cemented soil to reduce the harm caused by high compressibility. As strain increased, the slope of the stress drop section in small FA/HA specimens continued to grow until brittle damage occurred, followed by instantaneous stress reduction to zero after the peak. In Figure 4e, the stress of the cemented soil sample with FA/HA lower than 1 decreased rapidly after reaching the peak with increasing strain, and the total axial strain remained below 1.5%. The stress-strain curve at each curing time exhibited an overall increase over time, along with increased peak stress and brittleness. The greater proportion of fulvic acid led to a reduction in cement hydration products, resulting in a loose and uncompacted soil structure, ultimately altering the plastic properties of the cemented soil and enhancing its compressibility. 

In reality, the soil can be processed according to the actual working conditions. When the project requires high soil strength, the FA/HA value of the soil should be reduced, together with the curing time not being too long, so as to avoid brittle damage to the cemented soil. If the project requirements for soil strength is low, soil with high FA/HA can be selected as foundation soil, and its curing time should be appropriately extended.

#### 3.1.3. Deformation Characteristics

##### (1) Relationship between FA/HA and Failure Strain

Figure 5 illustrates the variation of failure strain with FA/HA and curing time. The figures reveal that an increase in FA/HA tended to raise the failure strain of the cemented soil as a whole, with the shorter-age specimens showing particularly obvious growth in failure strain. As curing time increased, the failure strain gradually decreased and stabilized in the later stage. Test results indicate that the effect of FA/HA variation on the failure strain of early cemented soil was more significant due to the more intense cement hydration reaction at early stages, making the impact of fulvic acid on hydration products more pronounced. The increase in the proportion of fulvic acid resulted in greater pore space within the cemented soil, reduced cementation strength between particles, a corresponding increase in failure strain, weakened resistance to deformation, and increased compressibility of the specimen.

The initial analysis of the figure data showed an exponential increase in damaging strain with increasing FA/HA, while it followed an ascending power trend with increasing curing time. A regression analysis was conducted to derive the relationship between damaging strain, FA/HA, and curing time, represented by Equation (2), which resulted in a correlation coefficient of 0.9323.
(2)$$\varepsilon_{f}=\frac{29.035}{T^{0.603}}-\frac{40.111}{T^{0.806}}\;\cdot \;0.772{}^{\gamma }$$
where *γ* is FA/HA, T is the curing time (d), and *T* = 7 d to 120 d.

##### (2) Relationship between FA/HA and Deformation Modulus E_50_

Figure 6 shows the variation of deformation modulus E_50_ with FA/HA. The figure indicates that the deformation modulus E_50_ decreased exponentially with the increase of FA/HA, while it gradually increased with the increase of curing time. This suggests that the effect of fulvic acid on the yield strength of the cemented soil structure was significant. The smaller the deformation modulus, the higher the compressibility of the soil. The increase of FA/HA may lead to the risk of greater deformation of cemented soil. The special structural characteristics of fulvic acid led to its absorption on the surface of cement and soil particles, which hindered the formation of cement hydration products and the interaction between soil particles and hydration products. The increase in fulvic acid content reduced the interparticle bowing and interparticle cementation strength within the cemented soil [25].

The correlation coefficient of 0.9846 was obtained by fitting the relationship between deformation modulus and FA/HA and curing time by regression analysis as shown in Equation (3). Deformation modulus is often used in the calculation of foundation deformation, and this formula can provide a reference for the foundation deformation estimation of sites with similar geological conditions.
(3)$$E_{50}=6.096\;\cdot {\;T}^{0.453}+33.067\;\cdot {\;T}^{0.397}\;\cdot \;0.555{}^{\gamma }$$
where *γ* is FA/HA, *T* is the curing time (d), and *T* = 7 d to 120 d.

### 3.2. Structural Analysis of Cemented Soil Based on Resistivity Theory

#### 3.2.1. Influence of FA/HA on Resistivity

Figure 7 displays the relationship curves of the initial resistivity of the cemented soil specimens at different curing times with the variation of FA/HA. Under the same curing conditions, the initial resistivity of the cemented soil sample continuously decreased with an increase in FA/HA. The initial resistivity reflects the characteristics of soil porosity and water content, among others. The increase in the proportion of fulvic acid hindered the hydration reaction of cement and affected the structure generation of cemented soil, leading to reduced water consumption, increased porosity, and decreased contact surface between soil particles, resulting in a decrease in resistivity. Soil samples with longer curing times exhibited reduced water content and soil porosity compared to early soil samples, which led to relatively higher resistivity.

From Figure 7, it can be observed that the initial resistivity exhibited an exponential relationship with the change in FA/HA. A regression analysis of FA/HA in exponential form with the initial resistivity *ρ_i_* was conducted to derive the two-factor equation for the initial resistivity *ρ_i_*, incorporating both FA/HA and curing time, represented by Equation (4). The correlation coefficient (R^2^) was 0.9970.
(4)$$\rho_{i}=4.65\;\cdot {\;T}^{0.52\;}+1.09\;\cdot {\;T}^{0.69}\;\cdot \;exp\left(\frac{-3.16\gamma }{lnT}\right)$$
where *ρ_i_* is the initial resistivity (Ω·m), *γ* is FA/HA, *T* is the curing time (d), and *T* = 7 d to 120 d.

#### 3.2.2. The Variation Pattern of Resistivity Structural Property Parameters

The quantification of soil microstructure using resistivity structural property parameters is primarily based on resistivity testing techniques, which measure the vertical and horizontal resistance changes of the soil during structural changes. This process allows obtaining two fundamental resistivity structural property parameters that reflect the internal structural characteristics of the soil: the average structure factor *F* and the average shape factor *f*. Analyzing the variation pattern of both parameters facilitates studying the quantitative changes in soil microstructure. *F* reflects the porosity and pore structure of the soil, while *f* reflects the particle shape of the soil. These parameters depend on the electrical conductivity, dielectric constant, particle shape, and frequency of the soil, regardless of the soil arrangement [22]. The resistivity structure characteristic parameters are calculated as follows [26]:(5)$$F=\frac{F_{V\;}{+2F}_{H}}{3}$$
(6)$$F_{V\;}=\frac{\rho_{V}}{\rho_{W}}$$
(7)$$F_{H}=\frac{\rho_{H}}{\rho_{W}}$$
(8)$$f=\frac{-logF}{logn}$$
where *ρ_V_* is the vertical resistivity (Ω·m), *ρ_H_* is the horizontal resistivity (Ω·m), *ρ_W_* is the resistivity of pore water (Ω·m), and its average value was 13.2 Ω·m measured by the test, and *n* is the porosity.

##### (1) Influence of FA/HA on Resistivity Structural Property Parameters

The average structure factor and average shape factor of the cemented soil at different curing times were calculated based on the resistivity data and other physical parameters of the cemented soil with different FA/HA. Figure 8 illustrates the relationship between resistivity structural property parameters and FA/HA.

From Figure 8, it is evident that both the average structure factor and the average shape factor of the cemented soil gradually decreased with the increase of FA/HA. Additionally, for cemented soil with different FA/HA values, both factors gradually increased with the increase of curing time, and their growth rate gradually decreased with the increase of FA/HA. It was observed that the growth rate of the average structure factor slightly decreased when the curing time exceeded 60 days. In contrast, the growth rate of the average shape factor significantly slowed down when the curing time exceeded 28 days. The longer the curing time, the more pronounced the effect of FA/HA variation on the average structure factor.

##### (2) Relationships between Unconfined Compressive Strength and Resistivity Structural Property Parameters

Correlation analyses were performed based on the results of unconfined compressive strength and resistivity structural property parameters. The relationships between *q* and *F*, and *q* and *f* for cemented soil with different FA/HA values were presented, respectively. As shown in Figure 9, a linear relationship was observed between the average structure factor and the unconfined compressive strength, represented by Equation (9). Additionally, a polynomial relationship was found between the average shape factor and the unconfined compressive strength, represented by Equation (10). Therefore, the unconfined compressive strength of cemented soil can be predicted based on the average structure factor and average shape factor.
(9)$$q=0.323F\;-\;0.087,\;R^{2}=0.857$$
(10)$$q=0.314\;-\;0.098f+0.135f^{2}{,\;R}^{2}=0.827$$
where *q* represents the unconfined compressive strength (MPa) of cemented soil with different FA/HA tested at each specified curing time, *F* represents the average structure factor, and *f* represents the average shape factor.

##### (3) Mechanism of Variation of Resistivity Structural Property Parameters

The analysis revealed a close relationship between the average structure factor, average shape factor, FA/HA, and curing time. The interdependent growth of unconfined compressive strength and resistivity structural property parameters characterizes the micro-mechanisms influenced by organic matter in cemented soil. Mixing cement with organic soil causes organic matter to envelop clay particles, hindering the reaction of the curing agent with clay particles [27]. The layer of absorption formed by the reaction between cement particles and fulvic acid hinders the hydration, crystallization, and cementation processes, preventing small particles from agglomerating into larger ones. Fulvic acid has a higher ability to consume calcium and aluminum compared to humic acid [12], leading to reduced soil particle agglomeration as its proportion increases. This phenomenon affects the pore structure and particle shape of the cement soil, as supported by the decrease in the average structure factor and average shape factor.

With increasing curing time, cement hydration and its interaction with clay particles are enhanced, resulting in increased cementitious strength of the structural unit and the agglomerate structure of the cemented soil. The hydrated crystalline material progressively fills the pores and fissures between soil agglomerates, forming a mesh structure between particles, ultimately reducing the pore space and forming a solid association. Longer curing time leads to an increased degree of structural cementation within the hydrated soil, stronger growth of microstructural linkage properties, and enhanced cementation ability between particles, resulting in an increase in the average structure factor and average shape factor.

## 4. Conclusions

This study examined the effects and mechanisms of fulvic acid and humic acid, components of organic matter, on the mechanical properties of cemented soil. It is committed to providing references for practical projects to select the appropriate curing time and FA/HA values based on the required strength and deformation. Based on the experimental results, the following conclusions are drawn: (1)As FA/HA increased, the uniaxial compressive strength and resistance to deformation of cemented soil with a certain amount of total organic matter gradually decreased, while its compressibility improved. The increase in curing time led to further development of the cement hydration reaction and an increase in strength. Fulvic acid had a greater impact on the strength and deformation properties of cemented soil compared to humic acid.(2)The initial resistivity and structural property parameters of the cemented soil decreased gradually with increasing FA/HA and increased gradually with age. The unconfined compressive strength and resistivity structural property parameters of the cemented soil exhibited similar growth characteristics. The presence of organic matter hindered cement hydration and limited the role of agglomeration. Fulvic acid had a more pronounced destructive effect on the cemented soil structure, and its decomposition disrupted the cement hydration products, resulting in a less compact soil pore structure, which affected inter-particle cementation and, consequently, the strength and deformation properties of the cemented soil.(3)The results of the unconfined compressive strength test were used to establish quantitative relationships between unconfined compressive strength, failure strain, deformation modulus, FA/HA, and curing time. The results of the resistivity test were used to establish quantitative relationships between unconfined compressive strength, resistivity, and structural property parameters. These relationships serve as the basis for designing soil mixing piles with organic matter.(4)In this paper, the organic matter with artificially prepared soil cannot consider the historical interaction of organic matter and soft soil, which will reduce the influence of organic matter on the curing effect of cement. At the same time, the research only studied the changing pattern of the mechanical and microscopic characteristics of the cemented soil with the change of FA/HA when the organic matter content is 10%. Therefore, quantitative equations are somewhat restrictive. The characteristics of cemented soil with different content and composition of organic matter need further study.

## Figures and Tables

**Figure 1 materials-16-05889-f001:**
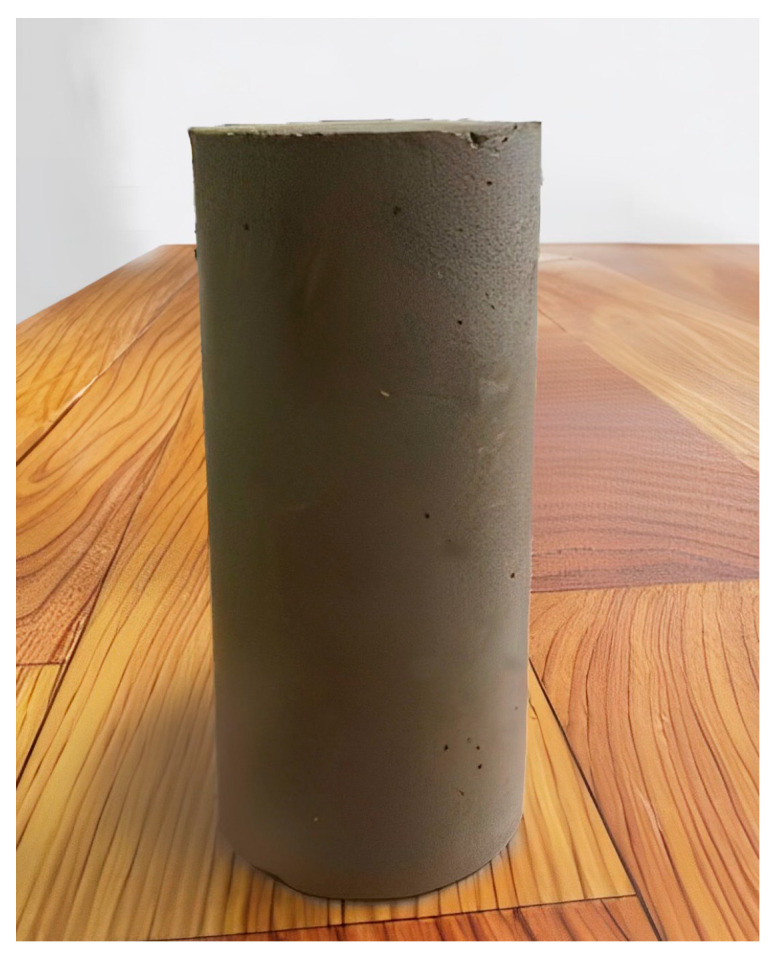
The molded sample of the test.

**Figure 2 materials-16-05889-f002:**
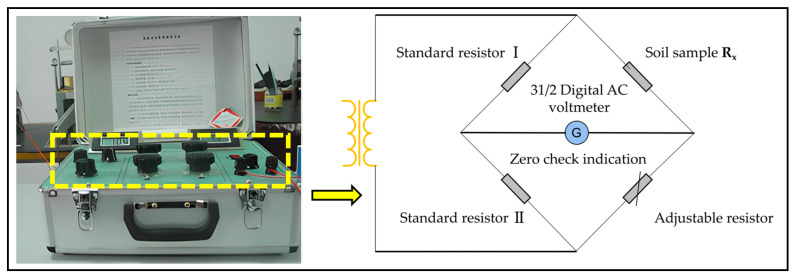
Low-frequency AC resistance tester and electrical schematic diagram.

**Figure 3 materials-16-05889-f003:**
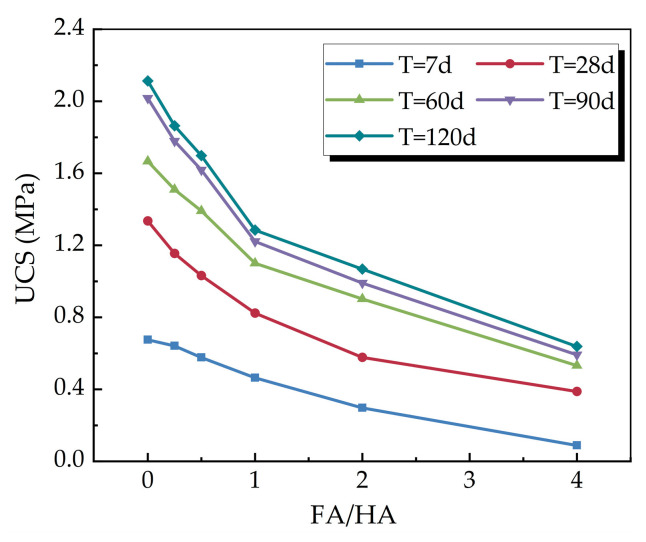
Relationship between unconfined compressive strength and FA/HA.

**Figure 4 materials-16-05889-f004:**
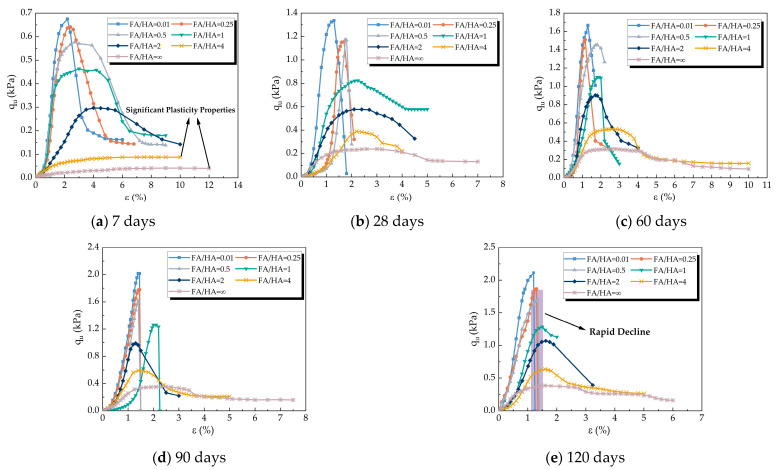
Stress-strain curves of cemented soil with different FA/HA at different curing time: (**a**) stress-strain curve at the age of 7 days; (**b**) stress-strain curve at the curing time of 28 days; (**c**) stress-strain curve at the curing time of 60 days; (**d**) stress-strain curve at the curing time of 90 days; (**e**) stress-strain curve at the curing time of 120 days.

**Figure 5 materials-16-05889-f005:**
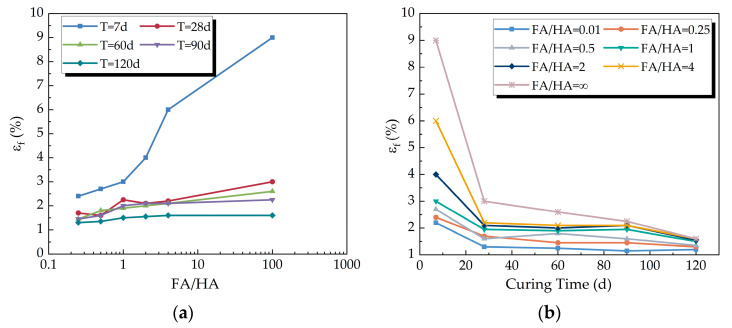
Variation of the failure strain with FA/HA and curing time: (**a**) variation of the failure strain with FA/HA; (**b**) variation of the failure strain with curing time.

**Figure 6 materials-16-05889-f006:**
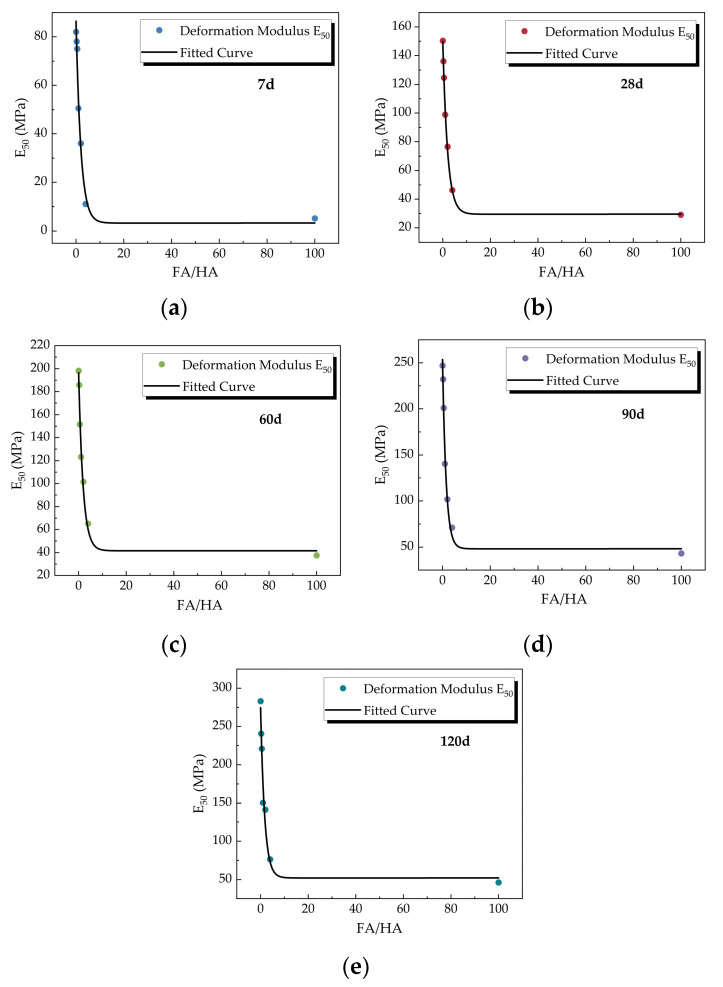
Relationship between deformation modulus E_50_ and FA/HA: (**a**) curing time *T* = 7 d; (**b**) curing time *T* = 28 d; (**c**) curing time *T* = 60 d; (**d**) curing time *T* = 90 d; (**e**) curing time *T* = 120 d.

**Figure 7 materials-16-05889-f007:**
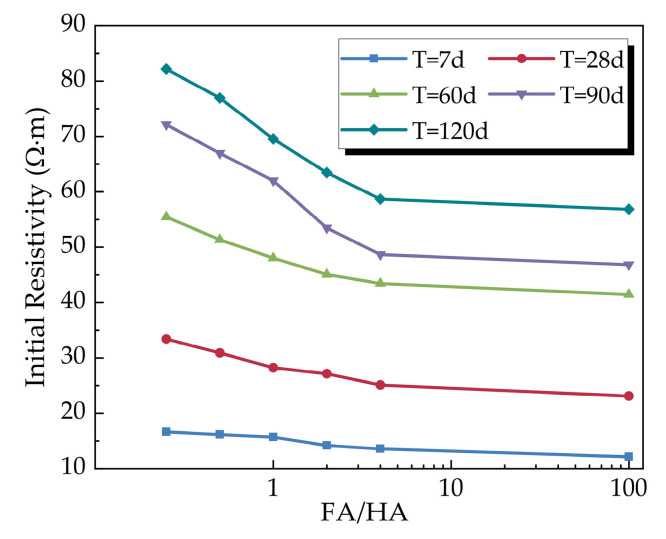
Variation of initial resistivity with FA/HA.

**Figure 8 materials-16-05889-f008:**
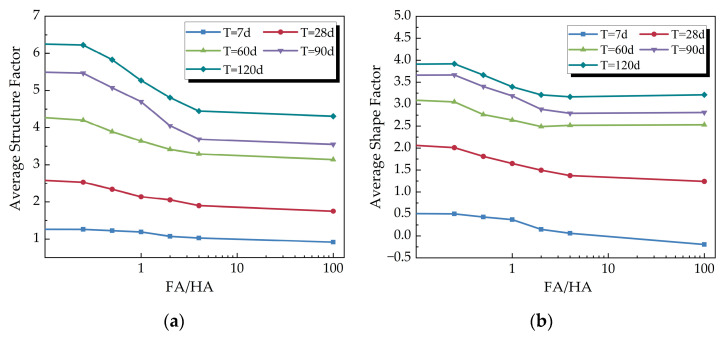
Variation of resistivity structural property parameters with FA/HA: (**a**) variation of the average structure factor with FA/HA; (**b**) variation of the average shape factor with curing time.

**Figure 9 materials-16-05889-f009:**
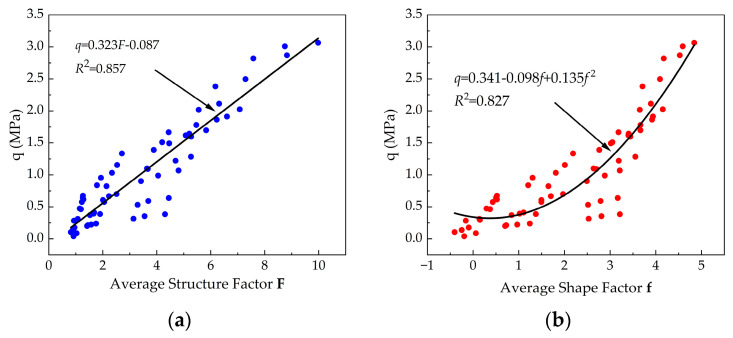
Relationships between the unconfined compressive strength and resistivity structural property parameters: (**a**) relationship between the unconfined compressive strength and the average structure factor; (**b**) relationship between the unconfined compressive strength and the average shape factor.

**Table 1 materials-16-05889-t001:** Main physical properties of the mucky clay.

Property	Value
Moisture content (%)	82.8
Density (g/cm^3^)	1.53
Specific gravity	2.72
Plastic limit	37.4
Liquid limit	75.7
Organic Matter (%)	2.55
pH	7.94

**Table 2 materials-16-05889-t002:** Main chemical composition of the cement.

Chemical Composition (%)	Value
CaO	49.18
MgO	1.62
SiO_2_	26.01
Al_2_O_3_	10.67
Fe_2_O_3_	2.83
K_2_O	0.95
SO_3_	3.76
TiO_2_	0.51
MnO	0.38
Loss of ignition	3.54

**Table 3 materials-16-05889-t003:** Test programs.

Test	FA/HA	Organic Matter Content (%)	Cement Content (%)	Moisture Content (%)	Curing Time (d)
Unconfined compressive strength test	0.01, 0.25, 0.5, 2, 4, ∞	10	15	80	7, 28, 60, 90, 120
Resistivity test					

Note: Moisture content is the mass of moisture as a percentage of the mass of dry soil; Cement content and organic matter content are their respective mass as a percentage of the mass of wet soil.

## Data Availability

Data can be obtained from corresponding author upon reasonable request.

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
