# Peer review of "Effect of Organic Matter Components on the Mechanical Properties of Cemented Soil"

_materials, 2023, doi:10.3390/ma16175889_

Round 1

Reviewer 1 Report

          In general, this manuscript is well prepared and easy to read. Since the test results were obtained from samples made with a single organic content of 10% only, the regression equations must be used with caution.

1.      All figures in the manuscript are somewhat difficult to read and must be improved with better resolution.

2.      Lines 174-179. The content of oxygen-containing functional groups in humic acid and fulvic acid should be discussed in the context of the chemical structure of FA and HA, so as to explain the difference in binding capability during cement hydration.

3.      Lines 245-249: The argument that “The special structural characteristics of fulvic acid led to its absorption on the surface of cement and soil particles, ……” ought to be brought up much earlier in the manuscript, not in this section on deformation modulus. 

Reviewer 2 Report

Review

Effect of Organic Matter Components on the Mechanical Properties of Cemented Soil

Abstract

Remarks:

Components present but may need minor writing for readability

The subject matter is original and important

Introduction:

Minor rewriting; the hypothesis is presented and is supported by the text

References are adequate

Remarks:

Always explain abbreviations when used for the first time.

However, the relationship between the main components of organic matter, mechanical properties, deformation characteristics of cemented soil, and their underlying microscopic mechanism has been rarely studied, and quantitative analysis is lacking. Considering the resistivity method’s superiority in studying the microstructure of stabilized soil, this paper aims to elucidate the microscopic mechanism by which the main organic matter components influence the mechanical and deformation properties of cemented soil. Organic substances, including fulvic acid and humic acid,

were incorporated into the test soil. The mixing ratio of fulvic acid and humic acid (hereinafter referred to as FA/HA) was varied to investigate their respective impacts on the mechanical and deformation properties of cemented soil through indoor experiments. Quantitative relationship equations between FA/HA and mechanical indices, electrical resistivity, and structural properties were established.

Remark:

The microscopic structure is studied by using the resistivity method. This needs to be highlighted the prevent a misleading interpretation

Materials and methods:

Adequately written, although writing could be polished; minor typos/grammar/

punctuation errors

The description of procedures needs minor clarification (clearly remediable)

The statical significance of data, presented in the tables, is not stated

Also, give some information concerning the accuracy of the measurements

Improve figure 1

Add a short explanation concerning the resistivity method to analyze the structure in the method part

Results:

Statical significance of findings not stated

Figures 2-8 (all) need to be improved

Conclusion

Statements and conclusions are presented but need minor revision to correlate with data and link with goals

Study implications or limitations are not presented, please add them to the paper

Remark:

Adding optical images of the microstructure could improve this paper

Adequately written, although writing could be polished; minor typos/grammar/

punctuation errors

The description of procedures needs minor clarification (clearly remediable)

Reviewer 3 Report

The manuscript presents interesting research on stabilized soils focusing on the influence of organic matter in various proportions on their properties. To improve the quality of the manuscript, the following comments should be considered:

Line 113: What does the # symbol next to the grade of cement mean?

Line 118: "0.5, 2, 4, and ∞". What does the symbol "∞" mean?

Line 123-124: "The organic content was fixed at 10%, and different FA/HA values were used to prepare the organic solutions” – this information is already included in lines 117-118.

Line 119-121 - this information shows that only cement grout with organic matter was tested. Where's the soil?

Line 125-126: "into Ф 5mm× 10mm molds" - are you sure these small dimensions? There should be a space between the number and the unit. What does the "Ф" symbol mean? Better to use text than symbols.

A photo of the molded sample in this form would be useful.

Line 127: "with 15% cement content and 80% moisture content" - this wording is unclear. Moisture means mixing water? 80% +15%=95%, and what is 5%? It is best to delete this sentence and present the exact recipe in table 3.

Line 130-131: "...and the specimens were further cured in the standard curing room until reaching the designated curing time.” - What temperature and humidity?

Table 3: Moisture content 0.80 means that the water in the cemented soil mix is 80% of the weight of the mix ingredients? This should be clarified.

2.3. Unconfined Compressive Strength Test - the described procedure related to limit deformations is taken from the standard? If so, please provide the number.

Figure 1: The diagram on the right is illegible. Sharpen the font.

Figure 2: Graph is unreadable - please sharpen it. There should be a space on the vertical axis between USC and (MPa).

Figure 3, 4, 5: Poor quality. It needs to be sharpened. Other drawings also require sharpening of the font.

Round 2

Reviewer 3 Report

My comments have been answered enough.